# PEER PRESSURE: MODEL-TO-MODEL REGULARIZATION FOR SINGLE SOURCE DOMAIN GENERALIZATION

## ABSTRACT

Neural networks are frequently deployed on multiple unseen target domains, which are distributionally different from the source domain on which the model is trained. Data augmentation is the most popular tool for single source domain generalization, which expands the source domain by generating simulated ones, commonly adopted by existing approaches. In this work, we observe that the performance of such augmentation-based methods in the target domains frequently fluctuates during training, posing challenges in model selection under realistic scenarios. We argue that the fluctuation stems from the inability of the model to accumulate the knowledge learned from diverse augmentations, exacerbating feature distortion during training. Based on this observation, we propose a novel generalization method, coined Parameter-Space Ensemble with Entropy Regularization (PEER), that uses a proxy model to learn the augmented data on behalf of the main model. The main model is updated by averaging its parameters with the proxy model, progressively accumulating knowledge over the training steps. Maximizing the mutual information between the output representations of the two models guides the learning process of the proxy model, mitigating feature distortion during training. Extensive experimental results demonstrate the effectiveness of PEER in reducing the OOD performance fluctuation and enhancing generalization across various datasets, including PACS, Digits, Office-Home, and VLCS. Notably, our method with simple random augmentation achieves state-of-the-art performance, surpassing prior approaches on sDG that utilize complex data augmentation strategies.

## 1 INTRODUCTION

Real-world deployment of deep neural networks frequently encounters domain shift, which refers to the discrepancy between the training domain and the unseen target domain on which the model is tested. An important aspect of domain shift is that it hinders the generalization of trained models (Kurakin et al., 2018). Nevertheless, a trained model is commonly expected to perform well on various OOD data, given a limited source of training data. Similarly, single source domain generalization (sDG) is the task of building a robust model that performs well across multiple OOD target domains, trained from a single source domain (Wang et al., 2021a). Existing approaches commonly utilize data augmentation to generate simulated target domains (Volpi et al., 2018b) and attempt to learn domain-invariant features from the augmented data.

This paper highlights an overlooked issue of leveraging data augmentation for sDG, particularly focusing on the fluctuation of OOD target domain performance amidst training, referred to as *mid-train OOD fluctuation* (Fig. 1). We find that this phenomenon stems from the model's incapability to accumulate the knowledge obtained from diverse augmentations and demonstrate that the features obtained from previous steps are largely distorted during training (see Fig. 2). We further illustrate that the fluctuation worsens when the

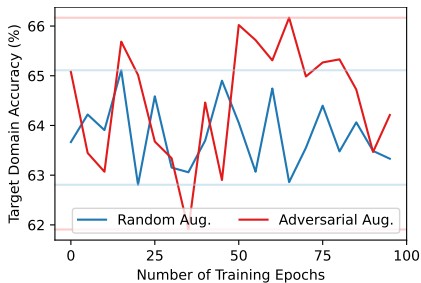

Figure 1: We observe that data augmentation improves generalization performance, but it causes fluctuations in target domain accuracy during the training. This phenomenon becomes more pronounced as the complexity of the augmentation increases.

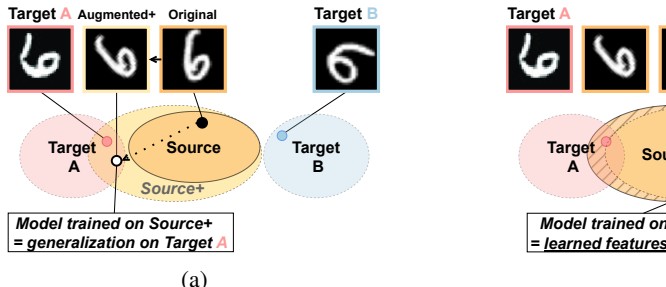

Figure 2: Illustration of pitfalls of augmentation in generalizing to unseen target domains. (a) Augmentation-based methods expand the source domain by providing diverse augmented samples (i.e., Source+). This enhances the model's generalization capability towards the unseen target domain (i.e., Target A). (b) Throughout the course of training, it iteratively simulates diverse unseen domains. However, at the same time, diverse augmentations lead to the distortion of the learned representations, thereby triggering OOD fluctuation.

model's trained features are distorted by augmented samples discrepant from the previously trained data and show that augmented samples are surprisingly inconsistent from their original state. This complicates model selection and potentially undermines generalization at test time, and thus, it is crucial to mitigate this issue.

Based on our observations, we suggest a novel generalization method coined PEER (Parameter-Space Ensemble with Entropy Regularization), that mitigates the augmentation-induced feature distortion by averaging parameters at various points along the model's learning trajectory (Izmailov et al., 2018). Specifically, our method leverages two interacting modules, i.e., the task model and the proxy model, to accumulate the knowledge acquired during training. The parameter-averaged task model guides the learning process of the proxy model, significantly reducing the aforementioned mid-train OOD fluctuation. Consequently, our framework stacks the generalization effect of varying data augmentation into the task model, reaching state-of-the-art performance across various sDG benchmarks (e.g., PACS, Digits), even in benchmarks where conventional sDG methods face difficulties in generalization (e.g., Office-Home, VLCS).

Our contributions are summarized as follows:

- We highlight an overlooked issue of the mid-train OOD fluctuation of augmentation-based sDG methods which poses serious issues in model selection and reveal that it stems from the distortion of the trained features.

- Based on our observation, we introduce PEER, a novel framework for sDG that stabilizes the learning process and boosts the target domain accuracy by accumulating the generalization effect of diverse augmentations using a parameter-space ensemble model.

- Our method achieves state-of-the-art performance across a wide range of benchmarks against existing augmentation-based sDG methods.

## 2 RELATED WORKS

**Domain generalization.** In the multi-source domain generalization (DG) literature, learning domain-invariant features has shown success in training robust models (Arjovsky et al., 2019). Specifically, these algorithms aim to disentangle the knowledge shared across domains (Klindt et al., 2021; Ren et al., 2021). A recent line of work highlighted the use of pre-trained models for model-to-model regularization, e.g., Cha et al. (2022) used an external pre-trained model to encourage the learning of domain-invariant features, and Li et al. (2023) expanded this approach by using multiple pre-trained models. In contrast, we refrain from using an external model and show that a training model can effectively perform regularization. On a different note, Arpit et al. (2022) studied the instability of the model's OOD performance and suggested an ensemble algorithm to alleviate the stochastic nature of the learning process. In contrast, we relieve the computational burden of ensembles by using a single

parameter-averaged model (Ainsworth et al., 2023; Rame et al., 2022; Jolicoeur-Martineau et al., 2023) and incorporate an alignment strategy (Choshen et al., 2022; Frankle et al., 2020) to assist this.

**Single source domain generalization.** In the sDG setting, only one domain is available for training, which makes it hard to apply conventional approaches developed for DG. To tackle this, a line of work focused on generating diverse domains using sophisticated data augmentation strategies, e.g., adversarial augmentation (Volpi et al., 2018b) or learnable augmentation modules (Fan et al., 2021; Qiao et al., 2020; Li et al., 2021; Wang et al., 2021b; Xu et al., 2023; Zheng et al., 2024). On the other hand, we reveal a universal phenomenon (i.e., mid-train OOD fluctuation) associated with utilizing data augmentation for generalization, and present a simple strategy to alleviate it.

**Mode connectivity and parameter-space ensembles.** Our work draws inspiration from the mode connectivity (Frankle et al., 2020) property of neural networks, which refers to the presence of a continuous manifold of non-increasing error that connects the minima identified by two global minimizers (i.e., trained models) (Garipov et al., 2018; Lubana et al., 2023). The concept is commonly used to justify how individual models can be merged to produce parameter-space ensembles (Wortsman et al., 2022; Rame et al., 2022) and also form the basis for designing model alignment methods to encourage mode connectivity between models (Entezari et al., 2021; Choshen et al., 2022; Ainsworth et al., 2023; Ramé et al., 2023). To analyze mode connectivity between models, a common practice is to measure the loss barrier (Frankle et al., 2020), quantified as the rise in loss values when the parameters of two models are averaged. Extending this, we suggest an effective alignment method to encourage mode connectivity between models trained with varying augmented data.

# 3 OBSERVATION: PITFALLS OF AUGMENTATION FOR GENERALIZATION

In this section, we reveal an overlooked problem in augmentation-based sDG methods. We first provide a brief background on the augmentation-based approaches to sDG (Sec. 3.1). Then, we highlight the performance fluctuation of models trained with data augmentation (Sec. 3.2).

## 3.1 AUGMENT-AND-ALIGN: AUGMENTATION-BASED APPROACHES TO SDG

Let $\mathcal{D}_S = \{(x_i, y_i)\}_{i=1}^N$ be a source domain where $x_i \in \mathcal{X}$ is an input image and $y_i \in \mathcal{Y}$ is its corresponding label. The goal of sDG is to build a model $F$ from $\mathcal{D}_S$ that is capable of generalizing to unknown target domains $\{\mathcal{D}_T^{(1)}, \cdots, \mathcal{D}_T^{(t)}\}$ distributionally different from the source domain. The model $F = C \circ H$ consists of a feature extractor $H : \mathcal{X} \to \mathcal{H}$ and the classifier $C : \mathcal{H} \to \mathcal{Y}$. Clearly, the classifier relying on the domain-specific features would not generalize to unseen target domains, and thus it is crucial to learn domain-invariant features from the source domain.

Existing approaches utilize data augmentation to simulate domain shift and aim to extract domain-invariant features by aligning the feature distribution between the original sample $x$ and its augmented view $\bar{x} = G(x)$, where $G$ is the augmentation function. The objective of such augmentation-based sDG approaches, omitting some arguments for simplicity, can be written as:

$$\arg\min_{H,C} \mathbb{E}_{(x,y)\in\mathcal{D}_S} \Big( \mathcal{L}_{\text{CE}}(C(H(x)), y) + \mathcal{L}_{\text{align}}(x, \bar{x}; H) \Big), \tag{1}$$

where $\mathcal{L}_{\text{CE}}$ is the cross-entropy loss and $\mathcal{L}_{\text{align}}$ is an alignment loss for capturing domain-invariant features by comparing $H(x)$ and $H(\bar{x})$. The commonly used alignment loss is InfoNCE (Oord et al., 2018), which lower bounds the mutual information $I(H(x), H(\bar{x}))$. Importantly, such alignment only guarantees to retrieve *augmentation-invariant* features (Von Kügelgen et al., 2021), and simple input transformations for generating the augmented views are often insufficient to capture *domain-invariant* ones (Aminbeidokhti et al., 2023). Therefore, recent methods devise more complex data augmentation strategies (Wang et al., 2021b; Li et al., 2021) to simulate diverse shifts in distribution.

However, it is still unclear whether such augmentation strategies can guarantee generalization to the target domain, especially given that it is unseen. In the sequel, we illustrate that this discrepancy makes the model performance fluctuate in the target domain.

Table 1: Empirical study of (a) target domain accuracy, (b) mid-train OOD fluctuation, and (c) source-target dataset distance. We use MNIST as a source.

| Method | SVHN | M-M | S-D | USPS | Avg. |
|---|---|---|---|---|---|
| (a) Target domain accuracy | | | | | |
| NoAug | 27.83 | 52.72 | 39.65 | 76.94 | 49.29 |
| RandAug [11] | 57.76 | 77.15 | 73.65 | 87.94 | 73.98 |
| AdvAug [38] | 62.21 | 82.20 | 69.39 | 85.26 | 74.77 |
| (b) Variance of the target domain accuracy | | | | | |
| NoAug | 4.76 | 2.77 | 1.72 | 0.32 | 1.33 |
| RandAug [11] | 2.51 | 1.04 | 1.05 | 1.49 | 1.52 |
| AdvAug [38] | 3.58 | 2.56 | 2.36 | 3.48 | 2.99 |
| (c) Source-target dataset distance [2] ($\times 10^3$) | | | | | |
| - | 3.46 | 2.65 | 2.75 | 0.92 | 2.45 |

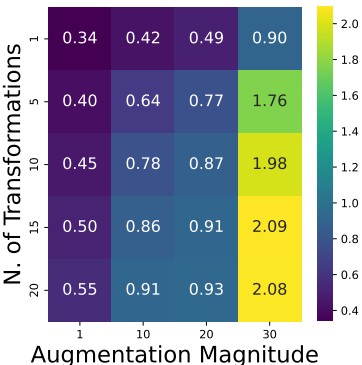

Figure 3: OTDD distance [2] between the original data (MNIST) and its augmented view.

## 3.2 Mid-train OOD fluctuation of Augmentation-Based sDG Methods

Recall Fig. 1, we find that augmentation-based sDG methods commonly exhibit large fluctuation of OOD performance throughout training, dubbed mid-train OOD fluctuation. Then, the following questions naturally arise: *"How does the fluctuation relate to the generalization performance? Where does the fluctuation stem from?"* Here, we investigate the relationships between the fluctuation and target domain accuracy through the lens of source-target dataset distance and examine the impact of data augmentation on the fluctuation.

We begin by observing that the target domain accuracy is closely related to the mid-train OOD fluctuation by comparing two augmentation-based sDG methods: random augmentation (RandAug, Cubuk et al. (2020)) and adversarial augmentation (AdvAug, Li et al. (2021)). As shown in the last column (Avg.) of Table 1-(a) and (b), the models with better generalization performance also display larger fluctuation. Clearly, the complexity of the data augmentation the models employed aligns with the target domain accuracy and fluctuation.

To further investigate their relationships, we adopt a similarity metric that measures the geometric distance between datasets (i.e., OTDD (Alvarez-Melis & Fusi, 2020)). By comparing different target domains (e.g., SVHN and USPS), we observe that the source-target discrepancy shown in Table 1-(c) is closely associated with the target domain accuracy and fluctuation. In other words, the models exhibit relatively small fluctuation on the target domain that is similar to the source domain (i.e., USPS) and vice versa (i.e., SVHN). Similarly, the models tend to show higher accuracy on target domains with smaller discrepancies (i.e., USPS) and vice versa (i.e., SVHN).

To better understand our observations above, we examine the discrepancy between the original dataset (MNIST) and its augmented view across varying degrees of random augmentation (Cubuk et al., 2020). As shown in Fig. 3, we observe that the discrepancy becomes more significant as the augmentation becomes diverse and its magnitude becomes stronger. Notably, such discrepancies often even exceed the source-target distance (i.e., 0.92 in Table 1-(c)).

Our observations suggest that data augmentation improves generalization capacity by simulating diverse domain shifts, but at the same time, it leads to the distortion of the learned representations and triggers mid-train OOD fluctuation, as depicted in Fig. 2. Based on our findings, we now proceed to present our method that effectively retains knowledge accumulated throughout the training, thereby alleviating fluctuations while achieving better generalization performance.

## 4 Parameter-Space Ensemble with Entropy Regularization

We now present a novel generalization method for sDG, coined Parameter-Space Ensemble with Entropy Regularization (PEER), that mitigates the augmentation-induced feature distortion and its associated issues (e.g., mid-train OOD fluctuation). Our approach involves two interacting modules with identical architectures: a frozen task model $F$ and a trainable proxy model $P$. The task model

guides the proxy model's learning process through entropy regularization of feature representations (Sec. 4.1). Subsequently, the task model is updated via parameter-averaging with the regularized proxy model, progressively accumulating the proxy model's knowledge throughout training (Sec. 4.2). The concept of our method is depicted in Fig. 4. The pseudo-code of our method is provided in Algorithm 1.

## 4.1 REGULATING THE PROXY MODEL WITH PEER

Our goal is to learn a robust task model $F$ from a single source domain that can generalize to multiple unseen target domains, where the task model consists of a frozen encoder $H_f : \mathcal{X} \to \mathcal{H}$ and a frozen classification head $C_f : \mathcal{H} \to \mathcal{Y}$, i.e., $F = C_f \circ H_f$. However, directly training the task model with varying augmented data is prone to feature distortion. Our key idea is to introduce a proxy model $P$ that trains on behalf of the task model and under the its guidance. Specifically, the proxy model $P = C_p \circ H_p$ shares the same architecture as the task model and consists of an encoder $H_p : \mathcal{X} \to \mathcal{H}$ and a classification head $C_p : \mathcal{H} \to \mathcal{Y}$. The proxy model is initialized by copying the task model at the beginning of training, i.e., $\theta_p \leftarrow \theta_f^{(0)}$ where $\theta_p$ is the parameters of the proxy model $P$ and $\theta_f^{(n)}$ is the parameters of the task model $F$ at $n$-th training epoch.

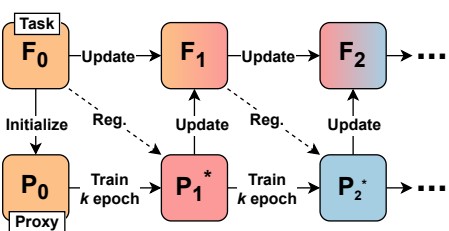

Figure 4: The PEER framework consists of two interacting modules: a proxy model $P$ and the task model $F$. During training, the task model retains the knowledge of the proxy model via parameter-averaging.

Our method PEER imposes regularization to the proxy model at the intermediate feature level. Instead of directly comparing the intermediate representation in $\mathcal{H}$, we map the representations from $H_f$ and $H_p$ using a shared projection head $R : \mathcal{H} \to \mathcal{R}$, following the empirical analysis by Gupta et al. (2022) and our experimental findings (Table 11) regarding its optimization efficacy.

The objective for PEER is then defined as:

$$\mathcal{L}_{\text{PEER}}(H_f(x), H_p(\bar{x})) = \text{BT}(R(H_f(x)), R(H_p(\bar{x}))), \tag{2}$$

where $x$ denotes the original sample and $\bar{x}$ the augmented view created by an augmentation function $G$, and BT (Barlow Twins) is a feature decorrelation loss (Zbontar et al., 2021):

$$\text{BT}(z, z^+) = \sum_i (1 - M_{ii})^2 + \lambda \sum_i \sum_{j \neq i} M_{ij}^2, \tag{3}$$

where $M$ refers to the cross-correlation matrix of the two feature representations $z, z^+$, and $\lambda$ is a balancing coefficient. The actual computation involves an empirical cross-correlation matrix $M$ between a batch of representations. The first term $\sum_i (1 - M_{ii})^2$ aligns two representations by spurring the diagonal values in $M$ of $(z, z^+)$ to be 1. The second term $\sum_i \sum_{j \neq i} M_{ij}^2$ minimizes redundancy in the representation by encouraging the off-diagonal values to be closer to 0. Intuitively, regularizing with PEER guides the proxy model $P$ to learn features selected by the task model $F$.

Notably, in Eq. (2), the task model and the proxy model receive nonidentical inputs $x$ and $\bar{x}$, respectively, reflecting our idea that the frozen task model is expected to provide a rich feature representation of the original sample $x$, while the training proxy model can better comprehend the newly augmented sample $\bar{x}$.

We train *only* the proxy model $P$ using a classification loss (i.e., cross-entropy) with the regularization:

$$\mathcal{L}_P = \sum_{x' \in \{x, \bar{x}\}} \mathcal{L}_{\text{CE}}(C_p(H_p(x')), y) + w \cdot \mathcal{L}_{\text{PEER}}(H_f(x), H_p(\bar{x})), \tag{4}$$

where $w$ is a balancing coefficient. In Sec. 4.3, we further elaborate the PEER regularization as a maximization of the mutual information (MI).

---

**Algorithm 1:** Parameter-space Ensemble with Entropy Regularization (PEER)

---

**1 Input:** Task model $F$ and its parameter $\theta_f$, augmentation function $G$, data from source domain $D_s$, augmentation reinitialization criteria $k$;

**2 Output:** Fully updated task model $F$ and its parameter $\theta_f$

**3** Pre-train $F$ with $D_s$ without $G$

**4** Initialize $P$ by setting its parameter $\theta_p$ with $\theta_f$ from $F$

**5** Initialize trajectory $\Theta \leftarrow \{\ \}$

**6 while** *not converge* **do**

**7**  **if** $n \% k$=0 **then**

**8**    Reinitialize $G$    // for random augmentation, change augmentation strength

**9**    $\Theta \leftarrow \Theta \cup \{\theta_p^{(n)}\}$  // save a snapshot of $P$

**10**    $\theta_f \leftarrow \text{AVERAGE}(\Theta)$ // update $F$ (Eq. (5))

**11**  **for** $i = 1 : n_{iterations}$ **do**

**12**    Augment the $i$-th mini-batch sampled from $D_s$ with augmentation function $G$

**13**    Train $P$ with PEER following Eq. (4)

---

## 4.2 ACCUMULATING KNOWLEDGE IN THE TASK MODEL WITH PEER

The task model $F$ is gradually updated through parameter-averaging with the proxy model $P$. This updating process progressively improves the task model's generalization throughout training, ensuring it remains effective as the regulator of the ever-growing proxy model (Burns et al., 2023). Specifically, we update the task model by parameter-averaging with the proxy model for every $k$ epoch through the proxy model's learning trajectory i.e., $\Theta = \{\theta_p^{(k)}, \theta_p^{(2k)} \cdots, \theta_p^{(\lfloor \frac{n}{k} \rfloor \cdot k)}\}$ where $n$ is the current training epoch, and update the task model with:

$$\theta_f \leftarrow \frac{1}{|\Theta|} \sum_{\theta \in \Theta} \theta. \tag{5}$$

Also, we reinitialize the augmentation function $G$ for every $k$ epoch (e.g., changing the policy – number and of transformations/ magnitude – of random augmentation). This periodic update of the task model allows it to stack the effect of diverse augmentations, similar to an ensemble model (Rame et al., 2022).

For the parameter-averaged task model to enjoy ensemble effects, it's crucial to ensure mode connectivity (Frankle et al., 2020) between the task model and the proxy model, which can be sufficed by sharing an identical initialization or backbone (Neyshabur et al., 2020). As our proxy model is initialized from the task model, it naturally satisfies this requirement. To further benefit parameter-averaging, the two models must be closely located in the feature space, which can be obtained by tuning the models on an identical source data (Ramé et al., 2023; Choshen et al., 2022). Our regularization with PEER (Eq. (2)) encourages the proxy model to be aligned with the task model in the feature space by treating the augmented domain similarly to the source domain. In Sec. 5, we show that the task model and the proxy model benefit from the regularization's alignment effect. In Appendix A, we empirically demonstrate that the task model cannot function as an effective regulator of the proxy model without the updating process (w/o ParamAvg. in Table 5).

## 4.3 DISCUSSION

**PEER as mutual information (MI) maximization.** The idea of PEER is that we can leverage the frozen task model to regularize the proxy model by maximizing the shared information between the two models. PEER aims to maximize the MI between the intermediate output features of the two encoders $H_f$ and $H_p$. The entropy regularization aligns the proxy model to the task model, preventing the proxy model from deviating too far from the task model. From this perspective, an intended objective for PEER could be formulated as $\max_H I(H_f(\bar{x}); H_p(x))$ where $I(X; Y) = \mathbb{E}_{p(x,y)}[\log p(x \mid y)/p(x)]$ indicates the mutual information (MI). PEER uses a feature decorrelation loss Eq. (3) (Zbontar et al., 2021) to maximize the lower bound of MI as a surrogate objective for MI optimization under a Gaussian assumption (Tsai et al., 2021). We further elaborate on the adequacy of feature decorrelation loss for MI optimization in Appendix A and report experimental results

of using an alternative objective e.g., InfoNCE (Oord et al., 2018) for Eq. (3) (Table 7). In Sec. 5, we provide experimental analysis on the effect of PEER by showing its effectiveness in alleviating augmentation-induced feature distortion.

# 5 EXPERIMENT

In this section, we investigate the following questions: (1) How effective is our method compared to prior sDG approaches? (Tables 2 and 3) (2) Does our method reduce the fluctuation of OOD performance? (Table 4) (3) What effect does our method have on the model's learned features and loss landscape connectivity? (Figs. 5, 6 and 7) (4) How effective is our method compared to previous model-to-model regularization approaches (Table 5) or ensemble methods (Table 6)?

## 5.1 EXPERIMENTAL SETUP

**Datasets.** Following prior works (Li et al., 2021; Wan et al., 2022), we evaluate our method on two standard benchmarks for sDG. **PACS** (Li et al., 2017) consists of 4 domains of differing styles (Photo, Art, Cartoon, and Sketch) with 7 classes. By default, we train our model with the Photo domain and evaluate it on the remaining target domains. **Digits** comprises of 5 different digit classification datasets, MNIST (Deng, 2012), SVHN (Netzer et al., 2011), MNIST-M (M-M) (Ganin et al., 2015), SYNDIGIT (S-D) (Ganin & Lempitsky, 2015), and USPS (Le Cun et al., 1989). We train our model with the first 10,000 samples of the MNIST dataset and assess its generalization accuracy across the remaining domains.

We also include Office-Home (Venkateswara et al., 2017) and VLCS (Fang et al., 2013), challenging benchmarks for sDG methods. **Office-Home** is a common multi-DG benchmark consisting of 4 datasets (Real-world, Art, Clipart, Product) with differing styles with 65 classes. We train on the Real-world domain and evaluate with the remaining domains. **VLCS** is also a benchmark for multi-DG, comprised of 4 datasets, PASCAL-VOC (V), LabelMe (L), Caltech-101 (C), and SUN09 (S) with varying styles. We used the PASCAL-VOC dataset as the source and the rest as target domains.

**Baselines.** We first consider ERM (Koltchinskii, 2011) and also compare our method with several strong augmentation-based approaches, i.e., M-ADA (Qiao et al., 2020), L2D (Wang et al., 2021b), PDEN (Li et al., 2021), and AdvST (Zheng et al., 2024). Finally, we include MetaCNN (Wan et al., 2022), which learns generalized meta-features.

**Implementation.** We use the same backbone architecture as prior works to ensure fair comparison. Specifically, we used AlexNet and multi-layer CNN for PACS and Digits, respectively, following earlier works (Wan et al., 2022; Li et al., 2021; Volpi et al., 2018a). For Office-Home and VLCS, we used ResNet-18. Additional experimental results across various backbone models (e.g., ResNet-18/50) are provided in Appendix (Tables 9 and 10). For the implementation of our method, we use random augmentation (Cubuk et al., 2020) to generate augmented samples. We set $k = 10$ and the balancing coefficients $\lambda = 0.005$, and $w = 2$ for all experiments. Hyperparameter studies are provided in Appendix D.1. We report the final test accuracy of the task model and report the OOD fluctuation measured as the variance of the target domain accuracy for every $k$-th epoch (Table 4). Throughout this section, we use the abbreviation RA for Random Augmentation and P for PEER.

## 5.2 MAIN RESULTS

In Tables 2 and 3, we report experimental results using the accuracy for each target domain and the mean accuracy across all target domains. In standard sDG benchmarks (i.e., PACS, Digits; Table 2), our method achieves state-of-the-art target domain accuracy in many of the target domains and outperforms all baselines in terms of mean accuracy. Notably, our method outperforms current SoTA methods by 2.30% and 0.96%. It is worth noting that our simple method boosted the mean accuracy of random augmentation (RandAug) by 7.08% ↑ in Digits and 3.76% ↑ in PACS.

In more challenging benchmarks (i.e., Office-Home, VLCS; Table 3), previous augmentation-based methods (e.g., PDEN, RandAug) show either small gains or negative effects in enhancing generalization. Similarly, naively applying random augmentation for these benchmarks lowered the target domain accuracy. In contrast, with PEER, the accuracy of the model trained with random augmentation shows a significant performance gain of 10.62% in Office-Home and 6.66% in VLCS.

Table 2: Target domain accuracy on PACS and Digits ($^\dagger$ indicates numbers are from original authors).

| | PACS | | | | Digits | | | | |
|---|---|---|---|---|---|---|---|---|---|
| Method | A | C | S | Avg. | SVHN | M-M | S-D | USPS | Avg. |
| ERM [31] | 54.43 | 42.74 | 42.02 | 46.39 | 27.83 | 52.72 | 39.65 | 76.94 | 49.29 |
| ADA$^\dagger$ [15] | 58.72 | 45.58 | 48.26 | 50.85 | 35.51 | 60.41 | 45.32 | 77.26 | 54.62 |
| M-ADA$^\dagger$ [46] | 58.96 | 44.09 | 49.96 | 51.00 | 42.55 | 67.94 | 48.95 | 78.53 | 59.49 |
| L2D$^\dagger$ [63] | 56.26 | 51.04 | 58.42 | 55.24 | 62.86 | 87.30 | 63.72 | 83.97 | 74.46 |
| PDEN [38] | 57.41 | 45.77 | 65.01 | 56.06 | 62.21 | 82.20 | 69.39 | 85.26 | 74.77 |
| AdvST [68] | 53.95 | 46.11 | 49.63 | 49.90 | 67.50 | 79.80 | 78.10 | 94.80 | 80.10 |
| MetaCNN$^\dagger$ [60] | 54.05 | **53.58** | 63.88 | 57.17 | 66.50 | **88.27** | 70.66 | 89.64 | 78.76 |
| RandAug [11] | 54.17 | 47.48 | 65.11 | 55.59 | 57.76 | 77.15 | 73.65 | 87.94 | 73.98 |
| PEER (ours) | **62.66** | 47.40 | **68.21** | **59.42** | **70.79** | 76.84 | **83.05** | **93.57** | **81.06** |

Table 3: Target domain accuracy on Office-Home and VLCS.

| | Office-Home | | | | VLCS | | | |
|---|---|---|---|---|---|---|---|---|
| Method | Art | Clipart | Product | Avg. | L | C | S | Avg. |
| ERM [31] | 52.78 | 40.19 | 68.73 | 53.90 | 59.06 | 97.30 | **74.25** | 76.87 |
| L2D [63] | 54.02 | 41.77 | 66.30 | 54.03 | 56.21 | 95.52 | 66.90 | 72.87 |
| PDEN [38] | 53.39 | 43.38 | 66.25 | 54.34 | 62.55 | 96.11 | 73.52 | 77.39 |
| RandAug [11] | 43.10 | 45.47 | 61.67 | 50.01 | 57.58 | 93.18 | 66.56 | 72.44 |
| PEER (ours) | **56.81** | **54.23** | **70.84** | **60.63** | **67.00** | **97.73** | 72.56 | **79.10** |

Table 4: Variance of the target domain accuracy.

| | PACS | | | | Digits | | | | | Office-Home | | | | VLCS | | | |
|---|---|---|---|---|---|---|---|---|---|---|---|---|---|---|---|---|---|
| Method | A | C | S | Avg. | SVHN | M-M | S-D | USPS | Avg. | Art | Clipart | Product | Avg. | L | C | S | Avg. |
| L2D [63] | 3.70 | 5.30 | 13.37 | 7.46 | 3.53 | 3.01 | 2.59 | 4.44 | 3.39 | 5.22 | 1.90 | 5.58 | 4.23 | 5.72 | 0.59 | 1.66 | 2.66 |
| PDEN [38] | 3.39 | 5.22 | 7.23 | 5.28 | 3.58 | 2.56 | 2.36 | 3.48 | 2.99 | 10.63 | 2.17 | 7.46 | 6.75 | 2.44 | 2.39 | 2.81 | 2.55 |
| RandAug [11] | 2.23 | 4.81 | 5.01 | 4.02 | 2.51 | **1.04** | 1.05 | 1.49 | 1.52 | **3.49** | 2.17 | 2.74 | 1.89 | 3.02 | 1.61 | **1.96** | 2.20 |
| PEER (ours) | **2.01** | **3.98** | **4.77** | **3.59** | **2.03** | 1.11 | **1.04** | **1.24** | **1.36** | 3.99 | **1.41** | **1.80** | **1.31** | **2.05** | 1.61 | 2.10 | **1.92** |
| Metric | Source-target dataset distance ($\times 10^3$) | | | | | | | | | | | | | | | | |
| OTDD [2] | 13.37 | 29.52 | 49.94 | 30.94 | 3.46 | 2.65 | 2.75 | 0.92 | 2.45 | 19.53 | 19.29 | 20.63 | 19.82 | 11.79 | 10.14 | 11.77 | 11.23 |

Finally, Table 4 demonstrates the fluctuation of OOD performance, measured as the variance across the target domain accuracy. We observe that our method successfully reduces the mid-train OOD fluctuation across all benchmarks. In our framework, the task model accumulates knowledge of the proxy model throughout the training. Thus, regularizing with the task model encourages the proxy model to preserve the knowledge of previous steps, similar to a memory buffer used in continual learning (Wang et al., 2024). In the next section, we illustrate that the task model indeed preserves the knowledge of the proxy model through parameter averaging.

### 5.3 DETAILED ANALYSIS ON PEER

#### 5.3.1 ADVANTAGES OF PEER IN MODEL-TO-MODEL REGULARIZATION

In Table 5, we demonstrate the advantages of PEER compared to previous approaches that utilize a pre-trained model (i.e., teacher) for regularization, where T+RA and P+RA refer to applying the teacher and the PEER regularization, respectively. We observe that both the teacher and the task model in PEER reduce the OOD fluctuation, while the fully-trained teacher (T+RA) often displays a stronger regularization effect compared to PEER (P+RA). However, PEER achieves superior sDG target domain accuracy in both datasets compared to the teacher. This is due to the teacher model's static nature, which limits its capability to process newly augmented samples. In contrast, our task model, evolving with the proxy model, is less vulnerable to these limitations.

We further validate the effectiveness of the updating process by ablating parameter-averaging (w/o ParamAvg. in Table 5). Instead of updating the task model by parameter-averaging, we simply freeze a snapshot of the proxy model for every $k$ epoch and use the latest snapshot as the regulator. As

Table 5: Comparitive study on PEER vs. Teacher.

| Method | Regulator | PACS | | | | Digits | | | | |
|--------|-----------|------|------|------|------|--------|------|------|------|------|
| | | A | C | S | Avg. | SVHN | M-M | S-D | USPS | Avg. |
| Variance of the target domain accuracy (OOD Fluctuation) | | | | | | | | | | |
| RandAug [11] | N/A | 2.23 | 4.81 | 5.01 | 4.02 | 2.51 | 1.04 | 1.05 | 1.49 | 1.52 |
| T+RA | Teacher | **1.27** | **2.49** | 5.30 | **3.02** | 1.95 | 1.17 | **1.10** | **1.11** | **1.33** |
| P+RA | PEER (w/o ParamAvg.) | 1.69 | 3.38 | **4.62** | 3.23 | **1.93** | 1.10 | 1.11 | 1.22 | 1.34 |
| P+RA | PEER | 2.01 | 3.98 | 4.77 | 3.59 | 2.03 | 1.11 | 1.04 | 1.24 | 1.36 |
| Target Domain Accuracy | | | | | | | | | | |
| RandAug [11] | N/A | 54.17 | **47.48** | 65.11 | 55.59 | 57.76 | 77.15 | 73.65 | 87.94 | 73.98 |
| T+RA | Teacher | 58.61 | 46.66 | 64.23 | 56.50 | 63.37 | 72.63 | 77.91 | 87.39 | 75.33 |
| P+RA | PEER (w/o ParamAvg.) | 57.73 | 46.69 | 61.33 | 55.25 | 59.99 | **77.26** | 72.3 | 88.28 | 74.46 |
| P+RA | PEER | **62.66** | 47.40 | **68.21** | **59.42** | **70.79** | 76.84 | **83.05** | **93.57** | **81.06** |

shown in Table 5, the non-averaged task model sacrifices the target domain accuracy for addressing OOD fluctuation, which illustrates the effectiveness of parameter-averaging.

### 5.3.2 EFFECT OF PEER ON PARAMETER-AVERAGING

Here, we investigate the effect of PEER regularization in benefiting parameter-averaging for the task model $F$ update. We observe that the regularization brings forth an alignment between different steps of the proxy model in its learning trajectory $\Theta$. To clarify, we find different steps of the proxy model $\theta_p^{(i)}, \theta_p^{(j)}$ to be aligned by the regularization. To show this, we follow the practice of Frankle et al. (2020) and analyze the loss barrier between snapshots of the proxy model in its learning trajectory. Fig. 5 illustrates the mode connectivity of the proxy model training with data augmentation with/without PEER on Digits (source: MNIST, target: SVHN). Here, we analyze the connectivity of the proxy model in its early stage of training ($\theta_p^{(0)}$) and at the late stage ($\theta_p^{(100)}$) by interpolating

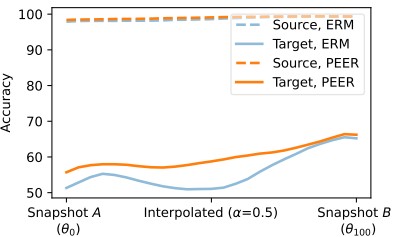

Figure 5: Mode connectivity in the proxy model's trajectory. PEER benefits parameter-averaging between snapshots of $P$ through its regularization effects.

the two $\alpha\theta_p^{(0)} + (1-\alpha)\theta_p^{(100)}$, where $\alpha \in [0,1]$ be the interpolation weight. We note that PEER aligns the model's snapshots($\theta_p^{(0)}, \theta_p^{(100)}$) in its learning trajectory, gifting a stronger performance gain when it is interpolated ($\alpha = 0.5$), especially in the OOD target domain. In other words, PEER's regularization enables the task model to function as a robust parameter-space ensemble, which can guide the proxy model's generalization to unseen target domains.

We further investigate the PEER's role in parameter-averaging in Table 6, specifically showing the failure cases of parameter-averaging without model alignment. Here, P-ENS refers to the parameter-space ensembles. In both PACS and Digits, parameter-space ensembling without regularization (P-ENS w/o PEER) falls behind ensembling with regularization. Notably in PACS, we observe failure cases of parameter-space ensembling without regularization, where the ensemble effect (i.e., gain in generalization ability) was very marginal. This failure case in parameter-averaging is an interesting observation as averaging the parameters between different training step snapshots of the same model has shown great success in many previous works (Grill et al., 2020; Izmailov et al., 2018). In Appendix C.2, we provide a deeper analysis of this topic.

### 5.3.3 EFFECT OF PEER ON LEARNED FEATURES

In this section, we analyze the PEER's effect on the learned feature representations. In detail, we share two results: (1) parameter-averaging allows the task model to accumulate the proxy model's knowledge, (2) the PEER regularization guides the proxy model to mitigate feature distortion (Appendix C.1).

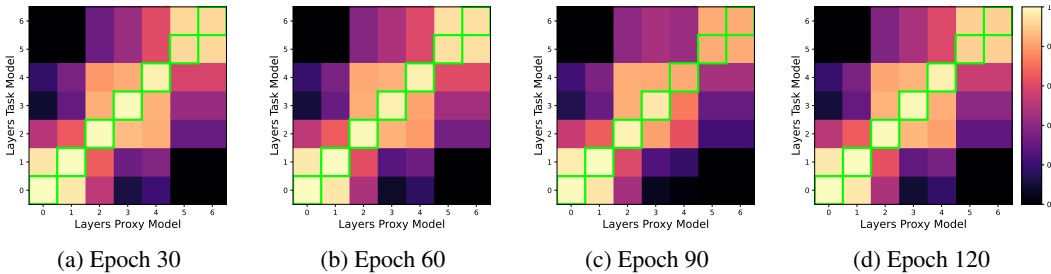

(a) Epoch 30      (b) Epoch 60      (c) Epoch 90      (d) Epoch 120

Figure 6: Layer-wise feature similarity between the fully updated task model and the proxy model at different epochs. The task model gradually accumulates the knowledge of the proxy model.

Table 6: The target domain accuracy of the parameter-space ensemble ($^\dagger$ indicates numbers are from original authors).

| Method | Ensemble | PACS | | | | Digits | | | | |
| --- | --- | --- | --- | --- | --- | --- | --- | --- | --- | --- |
| | | A | C | S | Avg. | SVHN | M-M | S-D | USPS | Avg. |
| ERM [31] | ✗ | 54.43 | 42.74 | 42.02 | 46.39 | 27.83 | 52.72 | 39.65 | 76.94 | 49.29 |
| MetaCNN$^\dagger$ [60] | ✗ | 54.05 | **53.58** | 63.88 | 57.17 | 66.50 | **88.27** | 70.66 | 89.64 | 78.76 |
| P-ENS w/o PEER | ✓ | **63.20** | 41.08 | 56.25 | 53.51 | **71.87** | 76.42 | 82.36 | 92.23 | 80.72 |
| P-ENS PEER (ours) | ✓ | 62.66 | 47.40 | **68.21** | **59.42** | 70.79 | 76.84 | **83.05** | **93.57** | **81.06** |

To show this, we follow the practice of Neyshabur et al. (2020) and compute the Centered Kernel Alignment (CKA) metric (Kornblith et al., 2019) between trained models. The CKA metric measures the similarity between feature representations, where $1.0$ indicates perfect alignment. Specifically, we compute and visualize the CKA similarity for different layers of the multi-layer CNN network trained on the Digits setting (see Appendix E.4 for details). Each matrix in Figs. 6 and 7 displays the similarity between the two models, its diagonal values indicating the similarity between corresponding layers' feature representations, i.e. brighter boxes indicate more shared knowledge.

We report that the parameter-averaging allows the task model to function similarly to a buffer which accumulates the knowledge of the proxy model across previous training steps. Fig. 6, we illustrate the feature similarity between the task model $F(\theta_f)$ and the proxy model $P(\theta_p)$. We can see that the fully updated task model is closely aligned with different stages of the proxy model's trajectory (indicated by bright diagonal values in Fig. 6), suggesting that the parameter-averaging effectively consolidates knowledge from various augmentations and preserves features that might otherwise be distorted during training. We continue this discussion on Appendix C.1, where we show that PEER plays an important role in addressing the feature distortion during training (Fig. 7).

### 5.4 ABLATION STUDY

We conduct an ablation study to evaluate the impact of various components on overall performance, including the regularization objective (Table 7), hyperparameters $w$, $\lambda$, and $k$ (Tables 8a and 8b) , model size (Tables 9 and 10) , and the role of the projection head (Table 11).

## 6 CONCLUSION

This paper presents PEER, a novel generalization method to address the issues of augmentation-based approaches to single source domain generalization. We highlight the feature distortion induced by augmentation, which triggers fluctuations in the target domain performance during training. Based on our observations, we propose a parameter-averaged task model that accumulates the generalization effect of the training proxy model. Entropy regularization on their learned feature representation aligns the two models, addressing feature distortion. Experiments on various datasets (PACS, Digits, Office-Home, VLCS) demonstrate the effectiveness of our method in stabilizing the learning process and enhancing the generalization performance.

**Reproducibility Statement**    For reproducibility, we provide the source code, the data pickle files, and the scripts used in our experiments. Please refer to the README.md file in the supplementary materials on how to access the datasets. We also used a fixed seed setting, which is implemented in the source code. We also include notebook (.ipynb) files to reproduce the figures appearing in our paper. Lastly, in Sec. 5.1 and Appendix E, we thoroughly explain how our method and its experiments are implemented.

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

# A  STUDY ON MODEL-TO-MODEL REGULARIZATION

In this section, we further study the topic of model-to-model regularization. We first begin by revisiting previous works on model-to-model regularization, highlighting the differences from our approach. Next, we provide experimental results on using a pre-trained teacher for regularization. Using this, we show the strength of our approach against previous model-to-model regularization methods.

**Previous Methods: Using a teacher for regularization.**  Model-to-model regularization is frequently used to boost a model's performance in tasks such as knowledge distillation (Hinton et al., 2015; Beyer et al., 2022) or generalization (Cha et al., 2022; Li et al., 2023). Here, an underlying idea is that the supervisor (i.e. teacher) be a model displaying strong performance, namely OOD robustness. A common approach is to use a pre-trained model that is trained on a large dataset, or with a larger model architecture. However, there exist issues in deploying strong teacher models for the sDG task. First, using pre-trained teacher models contradicts the grounding idea of single source domain generalization (sDG). To our understanding, the goal of sDG is to devise a generalization method that can function well in a realistic environment where the source data is limited. Reflecting this, the sDG setting strictly forbids the use of additional source domains for training. In this sense, using a model that is already trained on a much larger dataset seems to go against this. Furthermore, if the teacher model is available for use, a more efficient method would be to directly utilize the teacher for inference, while its operating cost would be much larger.

**Our Method: Using a group of PEER for regularization.**  Our approach to model-to-model regularization alleviates the irony of using a pre-trained teacher model by replacing it with a parameter-space ensemble (task model $F$). Unlike previous approaches (Cha et al., 2022; Li et al., 2023), the PEER does not violate the constraints of the sDG setting. Specifically, the task model in PEER does not use additional training data as it is the training model itself. Second, it is of an identical architecture to the training proxy model, hence we need not worry about excessive computation costs. Furthermore, using a task model regulator of the identical architecture allows the proxy model to directly update the task model via parameter-averaging, without additional cost. On the other hand, when using a pre-trained teacher model, updating the teacher would require excessive gradient computation (e.g., online distillation (Gou et al., 2021)).

More importantly, our approach to model-to-model regularization is more easily applicable to real-world problems than using a pre-trained teacher, owing to the adaptive nature of the task model. In PEER, the task model is created during the training process. Hence, the task model effortlessly adapts to the new dataset. This adaptivity makes PEER applicable to any given task or dataset. On the other hand, a teacher is a fixed model that is supposedly pre-trained on large datasets. The fixed nature of the teacher limits its applicability, as the teacher would only work if the teacher's pre-trained data is similar to the new training data. For instance, a strong digit classification (Deng, 2012) model will not function well as a teacher for the image classification task (Li et al., 2023).

**Experiment: PEER vs. Teacher**  In this section, provide detailed information on our experimental results reported in Sec. 5.3.1, and emphasize the competitiveness of PEER against using a strong teacher model for regularization. Specifically, we demonstrate that a task model in PEER serves as a more robust regulator compared to a pre-trained teacher model. Specifically, we empirically show that a suitable teacher model is not always available. For analysis, we use the PACS and Digits datasets and compare three model-to-model regularization methods (1) None: The baseline without model-to-model regularization (2) Teacher: Following the practice of Cha et al. (2022), we selected the pre-trained RegNetY-16GF (Radosavovic et al., 2020) as a teacher for PACS. In contrast, in Digits, we could not obtain a pre-trained model fit for use as the teacher. Hence, we follow the practice of Cha et al. (2022) and use a model pre-trained on both the source and target domains of Digits. We will later elaborate on why the RegNetY-16GF does not apply to the Digits experiment. (3) PEER: The task model in PEER has the same architecture as the proxy model. At the beginning of training, it is identical to the proxy model and then updated during the training process by averaging the parameters of the proxy model and the task model. The model is trained with random augmentation and follows the setup stated in Sec. 5.

We share the results of the experiment in Table 5. Here, the methods T+RA and P+RA refer to applying the teacher regularization and the PEER regularization, respectively. First, we compare the effectiveness of the two regulators (the teacher and the task model in PEER) in reducing the OOD target domain performance fluctuation. In Table 5, we see that both the teacher and the task model in PEER reduce the OOD fluctuation (measured as variance), while the teacher displays a stronger regularization effect than the task model. We view that this result reflects the reality that the teacher is a fully trained model, while the task model is updated alongside the proxy model's training process, and hence is a weak supervisor, at least at the beginning of training (Burns et al., 2023). On the other hand, we see that thePEER shows higher sDG target domain accuracy (59.42) in PACS than using a teacher (56.50). We believe that this results from the nature of the frozen teacher. To illustrate, the teacher is a frozen model, and hence a model regularized by the teacher may have been bound by the teacher's supervision. On the other hand, the PEER uses a task model that grows alongside the proxy model, and hence less likely to share the issues exhibited by the teacher. This pattern is repeated in the Digits experiment at Table 5, where the teacher was slightly better in reducing the fluctuation, while our method with PEER showed a higher target domain accuracy.

In Table 5, we also test the case when the task model is not updated with parameter-averaging i.e., PEER (w/o ParamAvg.). Instead of updating the task model via parameter-averaging, we simply froze a snapshot of the proxy model every $k$ epoch and used it as the regulator. Here, we can see that the non-averaged task model showed effectiveness in alleviating the OOD fluctuation while limiting the target domain accuracy.

We find that for certain tasks, a teacher model is hard to obtain. In other words, there is no universal model for use as the teacher. For instance, in the PACS experiment, the RegNetY-16GF displayed sufficient capabilities as a model-to-model regularize. However, using the RegNetY-16GF as the teacher for the Digits experiment was not available. Notably, RegNetY-16GF marked low validation accuracy in the target domain, nor was it able to guide the proxy model. We believe that this difference is derived from the discrepancy between the two datasets. For instance, PACS is a collection of images without any distortion, while Digits is a dataset solely comprised of digit images. Hence, we view that the large gap between the pre-trained dataset of the RegNetY-16GF and the Digit classification datasets is responsible for this behavior. This issue can be explained with the work of Wolpert & Macready (1997), where the authors demonstrate that there exists a trade-off between a model's performance on a certain task and the performance on all remaining tasks. In contrast, the PEER is applicable to any task, as it gradually adapts to the dataset using the proxy model.

# B    DISCUSSIONS

## B.1    DISCUSSION ON THE FLUCTUATION

We illustrate the mid-train OOD fluctuation in Fig. 1. Here, the worst-case performance of the fluctuating model (red) consistently falls below that of the stable model (blue). This describes the issues of deploying a fluctuating model, as the fluctuation poses challenges in early stopping and model selection.

Arpit et al. (2022) has studied a similar phenomenon within the multi-DG literature, attributing the fluctuation to the stochastic nature of the learning process (e.g., random seed, order of data). While we acknowledge the role of other contributing factors, we hypothesize that the mid-train OOD fluctuation primarily stems from the model's inability to accumulate the knowledge learned from varying augmentations. In specific, we view that the model's trained features are distorted, or forgotten during training (Kumar et al., 2022; Shi & Wang, 2024).

## B.2    DISCUSSION ON PEER AS A MUTUAL INFORMATION OPTIMIZATION

Here, we further elaborate on the PEER. Specifically, we elaborate on why optimizing with Eq. (3) can maximize the mutual information (MI) To recapitulate, the PEER aims to maximize the MI between the output feature representations of the task model $F$ and the proxy model $P$. However, directly optimizing MI is challenging, as its exact estimation is intractable (Paninski, 2003). There exists InfoNCE loss (Oord et al., 2018) which adopts a lower bound of MI (Poole et al., 2019) as a surrogate

Table 7: Target domain accuracy with different entropy regularization functions.

| Method | Reg. Obj. | PACS | | | | Digits | | | | |
|--------|-----------|------|------|------|------|--------|------|------|------|------|
| | | A | C | S | Avg. | SVHN | M-M | S-D | USPS | Avg. |
| PEER (ours) | BT [67] | **62.66** | 47.40 | **68.21** | **59.42** | **70.79** | **76.84** | **83.05** | 93.57 | **81.06** |
| PEER (ours) | InfoNCE [43] | 60.03 | **48.11** | 67.91 | 58.68 | 68.34 | 75.80 | 82.69 | **93.92** | 80.19 |

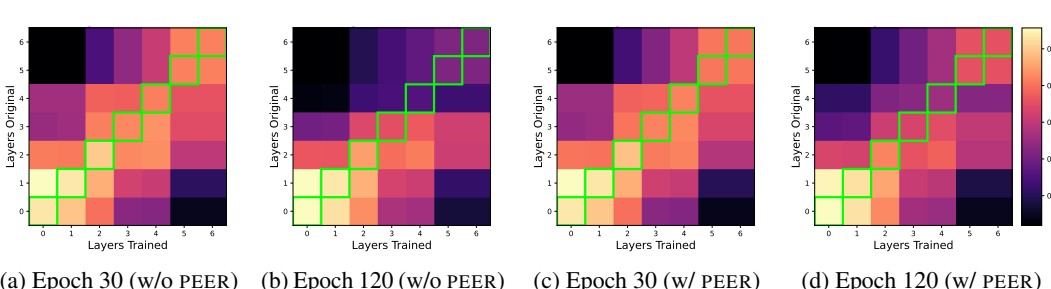

(a) Epoch 30 (w/o PEER)  (b) Epoch 120 (w/o PEER)  (c) Epoch 30 (w/ PEER)  (d) Epoch 120 (w/ PEER)

Figure 7: Layer-wise feature similarity (CKA) between the proxy model after initialization and after training with different epochs. Without PEER regularization, the model suffers feature distortion.

objective for MI optimization:

$$I_{\text{NCE}}(X;Y) \triangleq \mathbb{E}\Big[K^{-1} \sum_{i=1}^{K} \log \frac{\exp(f(x_i, y_i))}{K^{-1} \sum_{j=1}^{K} \exp(f(x_i, y_i))}\Big] \leq I(X;Y).$$

However, an issue of InfoNCE as a variational bound of MI is that InfoNCE requires a large batch size for convergence (Shrivastava et al., 2023; Hjelm et al., 2019), making it doubtful for use in small datasets (e.g., PACS). Consequently, we indirectly approximate InfoNCE with a feature decorrelation loss (Zbontar et al., 2021), based on empirical and theoretical results that show its functional proximity (Huang et al., 2021; Tao et al., 2022). Contrary to InfoNCE, the feature decorrelation converges effectively with small batch sizes and large vector dimensions, fit for many sDG settings with smaller datasets, or with images of large sizes.

In Table 7, we report the experimental results of replacing our regularization objective Eq. (3) with the InfoNCE. We find that both objectives are effective, while our default objective showed stronger results. We believe there are a number of factors behind this result (e.g., batch size, dataset (Balestriero et al., 2023)).

## C  EFFECT OF PEER ON THE MODEL

In this section, we further analyze the effect of PEER, namely on the proxy model's learned features and on its loss landscape.

### C.1  EFFECT ON LEARNED FEATURES (CONTINUED)

In this section, we study the effect of PEER on the learned feature representations.

We show that regularization plays an important role in reducing the proxy model's feature distortion during training. We compare two cases (a) *Without* PEER: CKA similarity of the proxy model $P$ at different epochs of training and its original state before training (b) *With* PEER: CKA similarity of the PEER applied proxy model $P$ at different epochs $n$ ($\theta_p^{(n)}$) and its original state ($\theta_p^{(0)}$). Notably, the diagonal elements in Fig. 7d are brighter in color than their counterparts (Fig. 7b), which indicates that PEER allows the proxy model to preserve its pre-trained features. The model is trained with random augmented MNIST data, and the feature similarity is also computed on the MNIST data.

Next, we provide a more detailed analysis. In Fig. 8, we report the case where there is no regularization from the task model (without PEER). Here, the diagonal values indicate the corresponding layers between the initialization and the trained model. We can see that as training continues (Fig. 7b), a

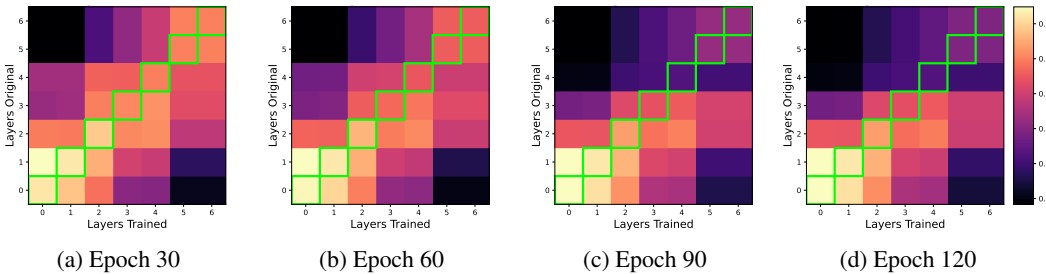

(a) Epoch 30  (b) Epoch 60  (c) Epoch 90  (d) Epoch 120

Figure 8: Layer-wise Feature Similarity (CKA) between the proxy model's initialization and the trained proxy model (without PEER). Without PEER regularization, the model suffers feature distortion.

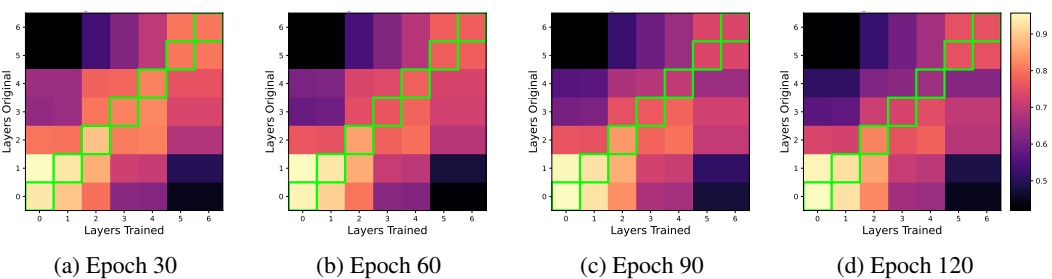

(a) Epoch 30  (b) Epoch 60  (c) Epoch 90  (d) Epoch 120

Figure 9: Layer-wise Feature Similarity (CKA) between the proxy model's initialization and the trained proxy model (with PEER). With PEER, the model suffers less feature distortion.

lot of trained knowledge is distorted in the later layers of the model. In contrast, Fig. 9 shows that when regularized with the task model (with PEER), the proxy model preserves a lot of knowledge even in the later epochs (Fig. 7d). Yet, we do not claim that PEER allows the proxy model to perfectly preserve its trained knowledge amidst diverse augmentation (Wolpert & Macready, 1997). Rather, we believe that by regularizing the proxy model, we can ultimately benefit the parameter-averaged task model. In the following section, we will empirically show that the regularization indeed benefits the parameter-averaging.

## C.2 EFFECT ON PARAMETER-AVERAGING (CONTINUED)

In this section, we provide an extended analysis of how regularizing the proxy model $P$ with the task model (i.e., PEER) aids parameter averaging. We argue that the regularization aids the ensembling effect by aligning different snapshots of the proxy model $\theta_p^{(i)}, \theta_p^{(j)}$ that were trained on very different augmented domains.

To show this, we perform a simple experiment: "Can parameter-averaging proxy model snapshots without regularization create a robust regulator?". Similar to PEER update, we periodically save snapshots of the proxy model training with random augmentation for every $k$ epoch. The experiment takes place in the PACS and the Digits benchmarks, and follows the same setting stated in Sec. 5. For PACS, the proxy model is trained for 200 epochs with random augmented data, where $k$ is set as 10. In Digits, the model is trained for 100 with $k$ set as 10. After training, we parameter average the saved snapshots to form a parameter-space ensemble. Note that in this case, no regularization took place.

We share the results in Table 6. As a recap, we explain the notations used in Table 6. In the table, P-ENS refers to the parameter-space ensembles. In both PACS and Digits, parameter-space ensembling with regularization (PEER) outcompetes ensembling without regularization (P-ENS w/o PEER). Notably in PACS, we observe failure cases of parameter-space ensembling without regularization, where the ensemble effect (i.e., gain in generalization ability) was very marginal. As noted in Sec. 5.3.2, this failure case is noteworthy since parameter averaging across different training snapshots of models with the same initialization has been highly successful in many prior studies (Grill et al., 2020; Izmailov et al., 2018).

Table 8: (a) Target domain accuracy and (b) fluctuation on PACS with different hyperparameters.

| | (a) Target domain accuracy | | | | | | (b) Variance of target domain accuracy | | | |
|---|---|---|---|---|---|---|---|---|---|---|
| Method | Hyperparam. | A | C | S | Avg. | Method | Hyperparam. | A | C | S | Avg. |
| | Hyperparameter: $w$ | | | | | | Hyperparameter: $w$ | | | | |
| Ours | w = 0.1 | 59.96 | 45.83 | 66.57 | 57.45 | Ours | w = 0.1 | 2.19 | 4.38 | 4.45 | 3.67 |
| Ours | w = 0.5 | 60.07 | 46.11 | 66.2 | 57.46 | Ours | w = 0.5 | 2.05 | 3.91 | 4.82 | 3.59 |
| Ours | w = 1.0 | 61.22 | 46.20 | 65.79 | 57.74 | Ours | w = 1.0 | 2.14 | 4.38 | 4.45 | 3.67 |
| Ours | w = 2.0 | 61.20 | 46.08 | 66.00 | 57.56 | Ours | w = 2.0 | 2.01 | 3.98 | 4.77 | 3.59 |
| Ours | w = 4.0 | 59.99 | 45.84 | 63.51 | 56.45 | Ours | w = 4.0 | 2.44 | 3.77 | 4.75 | 3.65 |
| Ours | w = 10.0 | 60.14 | 45.88 | 65.26 | 57.09 | Ours | w = 10.0 | 2.11 | 4.14 | 4.56 | 3.50 |
| | Hyperparameter: $\lambda$ | | | | | | Hyperparameter: $\lambda$ | | | | |
| Ours | $\lambda$ = 0.001 | 60.01 | 47.38 | 66.4 | 57.93 | Ours | $\lambda$ = 0.001 | 2.13 | 3.65 | 5.22 | 3.67 |
| Ours | $\lambda$ = 0.005 | 61.20 | 46.08 | 66.00 | 57.56 | Ours | $\lambda$ = 0.005 | 2.01 | 3.98 | 4.77 | 3.59 |
| Ours | $\lambda$ = 0.01 | 60.78 | 48.25 | 65.2 | 58.08 | Ours | $\lambda$ = 0.01 | 1.99 | 4.04 | 4.71 | 3.58 |
| Ours | $\lambda$ = 0.1 | 61.04 | 45.63 | 66.36 | 57.68 | Ours | $\lambda$ = 0.1 | 2.44 | 4.16 | 4.58 | 3.73 |
| | Hyperparameter: $k$ | | | | | | Hyperparameter: $k$ | | | | |
| Ours | $k$ = 1 | 56.99 | 42.30 | 67.25 | 55.51 | Ours | $k$ = 1 | 2.35 | 4.74 | 4.93 | 4.01 |
| Ours | $k$ = 5 | 62.17 | 47.42 | 63.52 | 57.70 | Ours | $k$ = 5 | 2.14 | 4.26 | 4.81 | 3.74 |
| Ours | $k$ = 10 | 61.20 | 46.08 | 66.00 | 57.76 | Ours | $k$ = 10 | 2.01 | 3.98 | 4.77 | 3.59 |
| Ours | $k$ = 20 | 63.45 | 47.11 | 62.23 | 57.60 | Ours | $k$ = 20 | 2.39 | 3.85 | 4.56 | 3.60 |

Generally, for a parameter-averaged model to display ensemble effects, some conditions should be simultaneously met (Ramé et al., 2023). (1) Share an identical initialization: models that share an initialization backbone tend to display very low loss barriers, showing mode connectivity. (2) Trained on same data: Models trained on identical source data (Choshen et al., 2022) tend to display mode connectivity, while models trained on varying data commonly do not (Ainsworth et al., 2023). In our case, the first condition is already met, while the second condition may have been broken due to the varying effects of data augmentation. Drawing from this, we hypothesize that the failure case above potentially derives from violating the second condition. In specific, we believe that the discrepancy between two very different augmented domains breaks the alignment between the model snapshots. In this sense, the PEER may help parameter-space ensembling by encouraging the regularized proxy model to align the newly augmented domain to the task model's source domain Sec. 4.1. Unfortunately, the alignment of models in its loss landscape is a topic that has not yet been thoroughly analyzed from a theoretical perspective, especially for models with deep architectures. While our empirical analysis may provide some insight, we believe further research is required on this topic.

# D    ABLATION STUDY

## D.1    STUDY OF HYPERPARAMETERS

We explore our method's sensitivity to hyperparameters. ($w$): $w$ is the hyperparameter used in Eq. (4), which functions as the balancing weight of the ERM objective and the regularization objective Eq. (2). We find that $w$ does not severely impact the course of training unless set to 0. We find that during training, the two losses are automatically tuned to match the magnitude of the $w$. ($\lambda$): $\lambda$ is the hyperparameter used for PEER that operates as the balancing weight of the two functions in Eq. (3). We begin with the value in the original paper (Zbontar et al., 2021) with $\lambda = 0.005$, and an alternate value $\frac{1}{r}$ introduced in Tsai et al. (2021) where $r$ is the length of a vector in $\mathcal{R}$ (regularization head output space). We observe that our method is resilient to the switch between two candidate values of $\lambda$ although we cannot guarantee they are optimal. ($k$): The augmentation reinitialization criteria $k$ is set as 10 for all experiments to ensure that the proxy model is sufficiently trained before switching the augmentation strategy. We find that switching $k$ with larger numbers causes no problem in training, but setting them too low $k < 2$ poses issues in aligning the proxy model with the task model, undermining the fluctuation stabilization effect.

We share the experimental results of our study on hyperparameters in Table 8a and Table 8b. As illustrated above, our method PEER showed resilience to changes in $w$ and $\lambda$. Both the target domain

Table 9: Target domain accuracy with different backbone architectures.

| Method | PACS | | | | Office-Home | | | | VLCS | | | |
|---|---|---|---|---|---|---|---|---|---|---|---|---|
| | A | C | S | Avg. | Art | Clipart | Product | Avg. | L | C | S | Avg. |
| | AlexNet | | | | ResNet-18 | | | | ResNet-18 | | | |
| RandAug [11] | 54.17 | **47.48** | 65.11 | 55.59 | 43.10 | 45.47 | 61.67 | 50.01 | 57.58 | 93.18 | 66.56 | 72.44 |
| PEER (ours) | **62.66** | 47.40 | **68.21** | **59.42** | **56.81** | **54.23** | **70.84** | **60.63** | **67.00** | **97.73** | **72.56** | **79.10** |
| | ResNet-18 | | | | ResNet-50 | | | | ResNet-50 | | | |
| RandAug [11] | 65.64 | 38.27 | 56.32 | 53.68 | 64.11 | 53.86 | 76.70 | 64.89 | 56.95 | 94.39 | 71.09 | 74.15 |
| PEER (ours) | **70.08** | **50.85** | **70.71** | **63.88** | **67.10** | **59.88** | **79.69** | **68.89** | **62.46** | **99.01** | **79.03** | **80.16** |

Table 10: Variance of the target domain accuracy with backbone architectures.

| Method | PACS | | | | Office-Home | | | | VLCS | | | |
|---|---|---|---|---|---|---|---|---|---|---|---|---|
| | A | C | S | Avg. | Art | Clipart | Product | Avg. | L | C | S | Avg. |
| | AlexNet | | | | ResNet-18 | | | | ResNet-18 | | | |
| RandAug [11] | 2.23 | 4.81 | 5.01 | 4.02 | **3.49** | 2.17 | 2.74 | 1.89 | 3.02 | 1.61 | **1.96** | 2.20 |
| PEER (ours) | **2.01** | **3.98** | **4.77** | **3.59** | 3.99 | **1.41** | **1.80** | **1.31** | **2.05** | 1.61 | 2.10 | **1.92** |
| | ResNet-18 | | | | ResNet-50 | | | | ResNet-50 | | | |
| RandAug [11] | 6.17 | 7.32 | **6.44** | 6.64 | 7.17 | **2.41** | 4.55 | 4.71 | 3.45 | 2.11 | **2.73** | 2.76 |
| PEER (ours) | **3.03** | **4.56** | 9.44 | **5.68** | **2.24** | 4.41 | **0.81** | **2.49** | **2.67** | **1.72** | 3.57 | **2.65** |

Table 11: Target domain accuracy with/without projection head $R$.

| Method | Proj. Head. | PACS | | | | Digits | | | | |
|---|---|---|---|---|---|---|---|---|---|---|
| | | A | C | S | Avg. | SVHN | M-M | S-D | USPS | Avg. |
| PEER (ours) | ✓ | 62.66 | **47.40** | **68.21** | **59.42** | 70.79 | 76.84 | **83.05** | **93.57** | **81.06** |
| PEER (ours) | ✗ | **62.76** | 43.26 | 66.00 | 57.34 | **76.34** | **93.07** | 68.96 | 80.36 | 79.68 |

accuracy and the OOD fluctuation were insensitive to the change in these two hyperparameters. However, we find that $k$ affects the fluctuation stabilization effect of our method, where setting $k < 1$ resulted in a slightly higher variance (4.01). This aligns with our expectations, as the proxy and task model may not benefit from the PEER regularization in just a single epoch. However, we discover that $k$ influences the stabilization of fluctuations in our method, with $k < 2$ leading to a slightly higher variance (4.01). This aligns with our expectations, as the proxy and task model may not fully benefit from the PEER regularization within a single epoch.

## D.2 STUDY OF MODEL SIZE

In this section, we present our findings on the effect of model size on generalization. We observe that larger models/backbones generally improve target domain accuracy. To demonstrate this, we replaced the backbones in three experiments: switching from AlexNet to ResNet-18 for PACS, and from ResNet-18 to ResNet-50 for Office-Home and VLCS. All backbones (AlexNet, ResNet-18, ResNet-50) were pre-trained on the same Imagenet1k dataset. We found that as the backbone size increased, target domain accuracy improved (Table 8a), though mid-train OOD fluctuation (variance of the target domain accuracy) increased slightly (Table 8b). However, the gain in accuracy outweighs the rise in variance, suggesting that larger models enhance generalization. We recommend future work to replace default backbones (e.g., AlexNet for PACS, 3-layer MLP for Digits) with larger ones (e.g., ResNets, ViTs).

## E IMPLEMENTATION DETAIL

In this section, we report the implementation details of our method.

### E.1 DATASETS

Here, we elaborate on the datasets used in our experiments.

**PACS** (Li et al., 2017) consists of 4 domains of differing styles (Photo, Art, Cartoon, and Sketch) with 7 classes. In default, we train our model with the Photo domain and evaluate the remaining target domains. We use the train/test split provided by the original paper (Li et al., 2017).

**Digits** is comprised of 5 different digit classification datasets, MNIST (Deng, 2012), SVHN (Netzer et al., 2011), MNIST-M (Ganin et al., 2015), SYNDIGIT (Ganin & Lempitsky, 2015), USPS (Le Cun et al., 1989). In our experiment, we train our model with the first 10,000 samples of the MNIST dataset and assess its generalization accuracy across the remaining four domains.

**Office-Home** (Venkateswara et al., 2017) is a common benchmark for DG, but not for sDG. The benchmark consists of 4 datasets (Real-world, Art, Clipart, Product) with differing styles with 65 classes. We train on the Real-world domain and evaluate the remaining domains.

**VLCS** (Fang et al., 2013) is also a common benchmark for DG, but not commonly used to evaluate sDG methods. The benchmark consists of 4 datasets (PASCAL-VOC, LabelMe, Caltech-101, SUN09) with differing styles with 5 classes. We train on the PASCAL-VOC domain and test the trained model on the remaining target domains.

For reproducibility, we provide the data used in our experiments as serialized pickle files (i.e., .pkl files).

### E.2 DATA AUGMENTATION

In our experiments, we used the Random Augmentation (Cubuk et al., 2020) strategy as the augmentation function. The random augmentation method has two hyperparameters, the augmentation magnitude, and the number of transformations. Generally, previous works have used random augmentation by fixing the hyperparameters.

As outlined in Algorithm 1, we periodically reinitialize the augmentation function by randomly selecting two hyperparameters, ensuring diverse augmented samples (Fig. 3). We find that changing the random augmentation configuration during training enhances generalization. While training a single model on these varied samples can lead to feature distortion, PEER mitigates this through parameter averaging. In Sec. 5, we have shown that simple random augmentation outperforms sophisticated augmentation strategies devised for single source domain generalization.

### E.3 BASELINES

Here, we provide detailed descriptions of each baseline. ERM (Koltchinskii, 2011) is the baseline of training without data augmentation, followed by several augmentation-based sDG methods that use complex adversarial schemes to generate challenging augmentations (Qiao et al., 2020; Wang et al., 2021b; Li et al., 2021). M-ADA (Qiao et al., 2020) adopted a Wasserstein autoencoder to regularize perturbation in the latent space, L2D (Wang et al., 2021b) takes a meta-learning approach to generate augmented domains, while PDEN (Li et al., 2021) and AdvST (Zheng et al., 2024) expand the training domains by progressively learning multiple augmentation modules, each simulating different domain shifts. Alternatively, MetaCNN (Wan et al., 2022) used a meta-convolutional network to learn generalized meta-features from local convolutional features. In contrast, we show that with PEER, simple random augmentation can outperform all the baselines.

### E.4 MODEL ARCHITECTURE

We report the details of model architectures used in our experiments. All models were built to match the architecture used in previous studies.

**Task Model**   The task model architecture varies in each experiment. For each experiment, we report the feature extractor $H$ and the regularization head $R$ of the task model $F$. Please note that the proxy model $P$ uses a model with an identical architecture as the task model $F$.

The task model used in the PACS experiment is AlexNet (Krizhevsky et al., 2012), pre-trained on ImageNet (Russakovsky et al., 2014). The model consists of 5 convolutional layers with channels of {96, 256, 384, 384, 256}, followed by two fully-connected layers of size 4096 units. The regularization head $R$ is a 3 layer MLP. The output dimension of the regularization head is 1024.

The task model used in the Digits experiment is a multi-layer CNN network (i.e. conv-pool-conv-pool-fc-fc-softmax). The architecture consists of two $5 \times 5$ convolutional layers, with 64 and 128 channels respectively. Each convolutional layer is followed by a MaxPooling layer ($2 \times 2$). The network also includes two fully connected layers with sizes of 1024, 1024 being the final output dimension of the feature extractor. The regularization head $R$ is a 2 layer MLP. The output dimension of the regularization head is 128.

Lastly, the task model used in the Office-Home and VLCS experiment is a ResNet-18 network. The ResNet is torchvision implemented and pre-trained on the ImageNet dataset. The regularization head $R$ is a 3 layer MLP. The output dimension of the regularization head is 1024.

**Teacher Model for the PEER vs. Teacher Experiment**   For the PEER vs. Teacher experiment, we used pre-trained models as a teacher model. In the PACS experiment, we used a pre-trained RegNetY-16GF model. The RegNetY-16GF is a variant of the RegNet family, a line of foundation image models introduced in Radosavovic et al. (2020) for image classification. The name of the model indicates its configurations, where the "Y" indicates the convolution method, and the "16GF" represents the model's capacity or complexity. We implement the model, and its model weights using the torchvision (Falbel, 2023) library. For the Digits experiment, we used a pre-trained model sharing the same architecture as the task model. As elaborated in Appendix A, this is because a pre-trained model fit for use in digit classification was hard to obtain. Hence, following the practice of Cha et al. (2022), we trained the model with the source and target domains of Digits to create an Oracle model.

### E.5   MODEL TRAINING

In this section, we elaborate on the details of the training process. We explicitly state the training hyperparameters (e.g., number of training epochs, augmentation reinitialization criteria $k$, learning rate, the type of the optimizer, learning rate scheduler, and batch size). All experiments are carried out using a single NVIDIA RTX 6000.

**PACS**   For the PACS experiment, we set the training epochs as 200, and the augmentation reinitialization criteria $k$ as 10. We tuned the number of epochs by analyzing the training behavior of the generators. We set the learning rate as $1e-4$, using the Adam optimizer (Kingma & Ba, 2015). The batch size was set as 128. In total, the PACS experiment took roughly 101 minutes.

**Digits**   For the Digits experiment, we set the training epochs as 1000, and the augmentation reinitialization criteria $k$ as 10. The learning rate was tuned as 0.0001, using the Adam optimizer. The batch size was set as 128. In total, the Digits experiment took roughly 233 minutes.

**Office-Home**   For the Office-Home experiment, the training epochs are set as 200, and the $k$ as 10. The learning rate was set as 0.0001, using the Adam optimizer. The batch size was set as 64. In total, the Office-Home experiment took roughly 128 minutes.

**VLCS**   Lastly, for the VLCS experiment, we train for 200 epochs, and the $k$ as 10. The learning rate was set as 0.0001, using the Adam optimizer. The batch size was set as 128. In total, the VLCS experiment took roughly 117 minutes.

### E.6   MODEL PRE-TRAINING

In this section, we report the information regarding the pre-training process. As mentioned above, we pre-trained our task model with the source domain before the main training procedure. We announce the number of pre-training epochs, the learning rate, the optimizer, the learning rate scheduler, and the batch size.

**PACS**  We pre-trained the AlexNet with the train data of the Photo domain, using the train split introduced in the original paper (Li et al., 2017). We pre-trained the model for 60 epochs, with a learning rate of 0.005 using the SGD optimizer. We further used the Step learning rate scheduler with a gamma rate (i.e. the strength of the learning rate decay) of 0.5. The batch size was set as 32.

**Digits**  For the Digits experiment, we set the number of pre-training epochs as 100, with a learning rate of 0.0001 using the Adam optimizer. The batch size was set as 256.

**Office-Home**  We pre-trained the ResNet18 with the train split of the Real World domain. We pre-trained the model for 100 epochs, with a learning rate of 0.0001 using the Adam optimizer. We used no learning rate scheduler. The batch size was set as 64.

**VLCS**  We pre-trained the ResNet18 with the train split of the PASCAL VOC domain. We pre-trained the model for 100 epochs, with a learning rate of 0.0001 using the Adam optimizer. We used no learning rate scheduler. The batch size was set as 64.

### E.7 HYPERPARAMETERS

In this part, we state the hyperparameters used in our experiments.

$\lambda$ is a balancing coefficient for $L_{\text{PEER}}$, an objective adopting the feature-decorrelation loss introduced in Zbontar et al. (2021). We tuned $\lambda$ using experimental results of the original paper and Tsai et al. (2021). In the original paper, the author reported the optimal value of the balancing term as 0.005, which remains consistent under varying projection dimensions. We set this as a starting point for hyperparameter tuning. We find that if $\lambda$ balances the off-diagonal term (i.e. redundancy reduction term) and the diagonal term (i.e. alignment term) to a similar degree, no significant differences are observed. Furthermore, switching $\lambda$ to $\frac{1}{d} \approx 0.0001$ showed no significant changes to the learning process. Here, $d$ denotes the projection dimension of the regularization head $\mathcal{R}$ (regularization head output space). While we cannot guarantee an optimal value for $\lambda$, we set $\lambda = 0.005$ for our experiments using PEER.

$k$ is an augmentation reinitialization criterion that performs two roles. (1) Augmentation reinitialization: For every $k$ epoch, the augmentation function is initialized. Here, reinitialization refers to the change in augmentation policy. For instance, for random augmentation, reinitialization refers to the change in augmentation strength. Alternatively, for augmentation techniques that utilize a learnable module (Li et al., 2021), the reinitialization would refer to reinitializing the parameters of the augmentation module. The motive behind the reinitialization is to expose the proxy model with diverse augmentations, (2) PEER update: For every $k$ epoch, the parameters of the proxy model $P$ are used to update the task model by averaging their parameters.

Lastly, $w$ is a hyperparameter used in Eq. (4), which balances the ERM objective and the regularization objective Eq. (2). As studied in Appendix D.1, $w$ does not affect the performance of our method. We have set $w$ as 2.0 based upon experimental results in Table 8.

