# OpenReview forum: "PEER Pressure: Model-to-Model Regularization for Single Source Domain Generalization"
_ICLR.cc/2025/Conference — ICLR 2025 Conference Withdrawn Submission_

### Official Review · Reviewer_vF9P · 2024-11-01

**Soundness:** 2
**Presentation:** 3
**Contribution:** 1
**Rating:** 3
**Confidence:** 5

**Summary:**

This paper argues that existing data augmentation-based DG methods frequently fluctuate during training, posing challenges in model selection under realistic scenarios. The authors argue that the fluctuation stems from the inability of the model to accumulate the knowledge learned from diverse augmentations, exacerbating feature distortion during training. Thus, they propose a generalization method, coined Parameter-Space Ensemble with Entropy Regularization (PEER), that uses a proxy model to learn the augmented data on behalf of the main model. The main model is updated by averaging its parameters with the proxy model, progressively accumulating knowledge over the training steps.

**Strengths:**

The authors show that the dissimilar the target domain is to the source domain, the larger the fluctuation. Also, the fluctuation becomes smaller by augmenting data to reduce the disimilarity.

**Weaknesses:**

- The novelty of the parameter-averaging methodology is limited. Is the proposed methodology specialized for domain generalization? How could it differentiate it from other ensemble methods?
- The author argues that the parameter-averaged task mode guides the proxy model's learning process, reducing the OOD fluctuation. However, the motivation behind the argument is not well explained. Is there any theoretical guarantee about this argument?
- For the OOD fluctuation, the authors investigate two augmentation methods. It may not be sufficient to demonstrate that every augmentation-based DG method has the same fluctuation problem.
- The details of the proposed parameter-averaging task are lacking. How do we choose the data augmentation methods for averaging? Are the different types of augmentation method-based proxy models average? Can we average the model parameter trained by adversarial training and geometric transformation?
- The experiments are not comprehensive.
    - Only small-scale datasets in DG are compared. Datasets such as Terra, NICO++, and DomainNets have not been evaluated.
    - Limited SOTA baselines are compared. Please see the references for suggested baselines.
    - The author only did the experiments on AlexNet and ResNet18. However, these backbones are quite outdated. How does the proposed method perform on ResNet50 or larger backbones? It is doubted whether the proposed method is scalable.
    - How does the proposed method perform with different types of advanced data augmentation-based DG methods?
    - Why did the author only evaluate three baselines on OfficeHome and VLCS?

References:
  - 2023 - CVPR - Simde: A Simple Domain Expansion Approach for Single-Source Domain Generalization
  - 2022 - TPAMI - Neuron Coverage-Guided Domain Generalization
  - 2021 - ICLR - Robust and Generalizable Visual Representation Learning via Random Convolutions

**Questions:**

Please see Weakness.

---

### Official Review · Reviewer_PrAJ · 2024-11-02

**Soundness:** 2
**Presentation:** 3
**Contribution:** 2
**Rating:** 6
**Confidence:** 3

**Summary:**

This paper proposes PEER PRESSURE, a regulariation method for single domain generalization. The authors claim the method can address the distortion problem caused by augmentation techniques and improve generalization power by introducing entropy regularization loss and a parameter space ensembling approach to update model weights. Furthermore, they show the superior performance of their method across different tasks and datasets compared to other methods and investigate the effectiveness of various components of their method.

**Strengths:**

* The paper is well-written and well-motivated.
* They investigate different components of their method thoroughly and in detail with different ablations.
* Their method is simple but effective in most cases.

**Weaknesses:**

* The novelty of their method is limited and incremental. They combined the Barlow twins [1] loss and teacher-student architecture methods (i.e., DINO [2])  in a supervised setting.
* The connection between the variance of target domain accuracy and generalizability is missing. The primary motivation of the paper is to decrease the augmentation distortion to improve the model generalization (Table 1, figure 3, and Table 5), which they measure with OOD fluctuation. However, in Table 5, a pretrained teacher and PEER without averaging have less OOD fluctuation yet lower accuracy than PEER with parameter averaging.

Minor Weakness:

* The numbers in all Tables don't have confidence intervals, so it is hard to grasp how significant the differences are. The authors should include confidence intervals or standard deviations from multiple runs.

**Questions:**

Suggestion:

* Since the method is based on Barlow twin loss, it would also be great to compare their method with Barlow twins.
* What would the effect be of using EMA (Earthmoving average) instead of parameter averaging?
* Does the method have the same effect on different augmentations except for random augmentation?

References:

1. Zbontar, J., Jing, L., Misra, I., LeCun, Y., & Deny, S. (2021, July). Barlow twins: Self-supervised learning via redundancy reduction. In International conference on machine learning (pp. 12310-12320). PMLR.
2. Caron, M., Touvron, H., Misra, I., Jégou, H., Mairal, J., Bojanowski, P., & Joulin, A. (2021). Emerging properties in self-supervised vision transformers. In Proceedings of the IEEE/CVF international conference on computer vision (pp. 9650-9660).

---

### Official Review · Reviewer_beht · 2024-11-03

**Soundness:** 2
**Presentation:** 3
**Contribution:** 2
**Rating:** 5
**Confidence:** 4

**Summary:**

This paper addresses the problem of single-source domain generalization, a practical challenge for deploying deep models in real-world applications. The authors focus on augmentation-based approaches and identify a performance fluctuation issue in models trained with strong augmentation techniques. To address this, they propose a parameter ensemble strategy that combines a sequence of historical models. Additionally, to mitigate feature distortion, a proxy model is introduced within a teacher-student learning framework, incorporating a Barlow Twins loss for decorrelation. Experimental results demonstrate the promise of the proposed method. However, I have several concerns, detailed as follows.

**Strengths:**

The studied problem is practical and interesting.

The overall findings are reasonable, and the proposed method appears technically sound.

Compared with RandAug, the method achieves significant improvements.

**Weaknesses:**

The paper still lacks key discussions and comparisons to clarify its originality and contributions. Please refer to my questions for detailed comments.

**Questions:**

The paper overall gives the impression that it is primarily an exploration of the effects of self-supervised learning (SSL) in domain generalization. This is due to the similarities between the proposed PEER teacher-student framework and BYOL, where BYOL uses EMA while PEER employs parameter ensemble (here, I am also interested in a comparison between EMA and parameter ensemble). Additionally, the Barlow Twins loss, widely used in SSL, is integrated as well. If this interpretation is accurate, a comprehensive discussion with SSL methods (e.g., BYOL, Barlow Twins) and SSL in the context of domain generalization or model robustness (e.g., works [A, B, C]) would be valuable. However, the paper currently lacks these discussions and comparisons with these methods.

Although the proposed method is augmentation-based, I believe the comparisons should not be limited to augmentation-based methods alone. It is also essential to compare with other advanced domain generalization methods, such as [D, E]. Currently, the only non-augmentation-based method included in the comparison is MetaCNN, published in 2022.

All experiments and findings in this paper are based on convolutional models. It is unclear whether these results would hold for other types of deep models, such as Vision Transformers, ResNeXt, Vision Mamba, and others. Demonstrating that the findings extend to a broader range of models would significantly enhance the work’s generality.

Since augmentation-based methods are frequently used to assess model robustness, such as on out-of-distribution target data with corruptions in the ImageNet-C dataset, including comparisons on ImageNet-C would strengthen the empirical results of the paper a lot.

What would the performance be if PEER did not alter the augmentation policy every $k$ epochs—specifically, the number and magnitude of transformations in random augmentation—by using the same random augmentation strategy throughout all training epochs?

Comparisons with RandAug using the proposed parameter ensemble strategy are encouraged to be included in Table 5.

I would be pleased to raise the score if the authors could address these questions.

[A] Selfreg: Self-supervised contrastive regularization for domain generalization

[B] PCL: Proxy-based Contrastive Learning for Domain Generalization

[C] CbDA: Contrastive-Based Data Augmentation for Domain Generalization

[D] Test-time classifier adjustment module for model-agnostic domain generalization

[E] Towards Unsupervised Domain Generalization

---

### Official Review · Reviewer_2XQM · 2024-11-04

**Soundness:** 2
**Presentation:** 2
**Contribution:** 1
**Rating:** 3
**Confidence:** 4

**Summary:**

This paper argues that existing augmentation-based approaches to sDG struggle with target domain performance fluctuations during training, primarily due to an inability to retain accumulated knowledge from diverse augmentations. To address this, the authors introduce PEER, which leverages a proxy model to learn augmented data while progressively updating a main model by averaging its parameters with the proxy. The experimental evaluation demonstrates that PEER outperforms baseline methods across several domain generalization benchmarks.

**Strengths:**

The overall structure is logical, with a clear flow from the identification of the problem to the proposed solution and subsequent empirical validation. The experimental setup and baselines are well described.

**Weaknesses:**

1. The core idea of parameter averaging is not new, and the contribution may not be seen as significantly advancing beyond existing ensemble and regularization techniques.
2. The authors argue that data augmentation leads to feature distortion. However, it is unclear what feature distortion means, and its existence is not substantiated by theoretical or experimental analysis.
3. The authors also argue that better generalization correlates with larger fluctuation during training. However, the empirical results shown in Table 1 only compare three methods and do not fully support the argument.
4. The baselines compared against are somewhat dated, and it would be beneficial to compare with more recent sDG methods. In addition, the proposed method should be compared to existing parameter averaging methods.
5. In Table 8, it seems the proposed regularization technique does not affect the performance much.

**Questions:**

See above.

---

### Note · Authors · 2024-11-13

I have read and agree with the venue's withdrawal policy on behalf of myself and my co-authors.